# Clinical Observations Associated with Phenobarbital Serum Monitoring to Manage Epilepsy in a California Sea Lion with Domoic Acid Toxicosis

Claire A. Simeone [1,2,*], Gregory Scott [2], Ryan A. Navarro [3] and Diana Procter [2]

1   Sea Change Health, Sunnyvale, CA 94086, USA
2   Six Flags Discovery Kingdom, Vallejo, CA 94589, USA; gscott@sftp.com (G.S.); dprocter@sftp.com (D.P.)
3   School of Veterinary Medicine, St. George's University, Grenada FZ818, West Indies; rnavarro@sgu.edu
*   Correspondence: claire@seachangehealth.org

**Abstract:** The marine algal toxin domoic acid is an important threat to marine mammal health, and exposure can lead to both acute neurologic signs and a chronic epileptic syndrome in California sea lions (*Zalophus californianus*). Phenobarbital has been used for several decades to manage seizures, although reports are limited correlating dosing, serum monitoring and clinical efficacy in this species. This report details serum monitoring over 33 months in an 8-year-old male sea lion. Seizure control was achieved when phenobarbital concentrations were above 18 μg/mL, and sedation and ataxia were noted when concentrations were above 35 μg/mL. There was no clinically significant difference between phenobarbital concentrations resulting from once-daily versus twice-daily dosing. Serum levels remained detectable as far as 101 days after administration, and remained stable during periods of prolonged anorexia, although dramatic decreases in serum concentrations were noted immediately after normal eating resumed. For this animal, a serum phenobarbital target range of 20–30 μg/mL was achievable with a dose of 1.5 mg/kg once daily followed by therapeutic monitoring, and this is a reasonable recommended concentration and initial dose for clinicians treating this species. Long-term seizure control may be difficult to achieve with anti-epileptic drugs such as phenobarbital alone, and further research is needed to make novel options useful for clinical management of biotoxin-related neurologic disease in this aquatic species.

**Keywords:** *Zalophus*; biotoxin; epilepsy; anti-epileptic drug; health

## 1. Introduction

Exposure to the marine algal toxin domoic acid (DA) is the most common cause of neurological abnormalities in California sea lions (*Zalophus californianus;* CSL) that strand along the coast of California, USA, and it is an important threat to marine mammal health [1]. As harmful algal blooms are increasing in frequency and intensity under changing ocean conditions, the resulting DA toxicosis has become increasingly common since it was first documented in this species in 1998 [2–5]. Acute neurologic signs can include ataxia, seizures and coma [6], while chronic, previous sublethal exposure to the toxin can result in a chronic epileptic syndrome [7]. Animals that exhibit these neurologic signs of DA toxicosis may strand on beaches and be rescued for rehabilitation and treatment. Depending on the clinical course of the disease, outcome options include release back into the wild, humane euthanasia, or animals may be deemed non-releasable by NOAA National Marine Fisheries Service and placed in a long-term care facility [8]. For animals with chronic epilepsy, seizures are controlled with anti-epileptic drugs (AEDs).

Phenobarbital (PB) has the longest history of chronic use of all AEDs in veterinary medicine and has long been the AED of choice for seizures with non-metabolic etiology in dogs and cats due to its low cost and low incidence of adverse effects [9–11]. PB has been used for several decades to treat seizures in CSL both in wild animals with acute

toxicosis in rehabilitation and in epileptic animals placed in long-term care facilities, but peer-reviewed reports correlating dosing, PB serum concentrations and clinical efficacy are limited in this species (Table 1). This report details long-term PB administration and serum PB monitoring in a single CSL with chronic epilepsy secondary to DA toxicosis.

**Table 1.** Published oral phenobarbital dosing and serum phenobarbital concentrations in California sea lions.

| Reference | Dose (PO) | Reported PB Serum Concentration [a] | Number of Animals |
|---|---|---|---|
| *Current study recommendations* | *1.5 mg/kg SID* | *20–30 µg/mL* | *1* |
| Iwata 1998 [12] | 2.0 mg/kg BID | 14.1–20.6 µg/dL | 1 |
| Gage 1999 [13] | 1.0–1.5 mg/kg SID-BID | NR | 1 |
| Gulland et al. 2002 [6] | 2.0 mg/kg | NR | NR |
| | 4.0 mg/kg BID for 2d, then 2.0 mg/kg BID for 5d | NR | NR |
| Field et al. 2012 [14] | 1.0 mg/kg | NR | 1 |
| Haulena et al. 2003 [15] | 2.0 mg/kg BID | 10–30 µg/L | 3 |
| | 4.0 mg/kg BID | >30 µg/L | 3 |
| | 6.0 mg/kg BID | >30 µg/L | 3 |
| Dold et al. 2005 [16] | 4.0 mg/kg SID | NR | 1 |

PO = per os. SID = once daily administration. BID = twice daily administration. PB = phenobarbital. d = day. NR = not reported. [a] Note differences in units reported for PB concentrations.

## 2. Materials and Methods

### 2.1. Case

This case describes an 8-year-old castrated male CSL. The sea lion stranded along the central coast of California in 2017, and three attempts at rehabilitation and release were attempted but were unsuccessful. A generalized convulsive seizure was observed while in rehabilitation. Veterinarians deemed the sea lion unlikely to survive in the wild, and it was transferred to a facility for permanent care. An initial MRI scan obtained in January 2018 showed no structural abnormalities, but a second MRI study obtained in October 2018 showed evidence of unilateral hippocampal atrophy. The animal was diagnosed with epilepsy caused by suspected DA toxicosis based on observed seizure-like activity, prolonged anorexia, radiologic evidence of hippocampal atrophy [17] and absence of infectious disease agents. To control seizures, AEDs were administered throughout the study period of June 2018 to February 2022. Oral medications were placed into the branchial cavity of a Pacific herring (*Clupea harangus*) and administered after acceptance of a non-medicated herring to reduce the chance of medication rejection. PB doses ranged from 0.7 to 3.4 mg/kg per os (PO), either once a day (SID) or twice a day (BID). Diazepam was administered at a dose of 0.05–0.1 mg/kg PO SID-BID, primarily for its behavior-modulating effects as opposed to its anticonvulsant activity. Levetiracetam was administered alone at 32 mg/kg PO BID for 8 weeks but was discontinued due to inconsistent administration with anorexia. Lorazepam (0.1–0.2 mg/kg intramuscular (IM)) was administered when a seizure was observed. In October 2020, a xenotransplantation procedure was performed to deliver interneuron progenitor cells into the damaged hippocampus [18].

### 2.2. Medical Record Review

Veterinary medical records were reviewed from June 2018 to February 2022. Body weight, frequency of observed seizures, seizure phenotype, dosage of PB, serum PB concentration, timing of blood sample in relation to PB administration, number of days after last PB dose adjustment, dosage of other AEDs, and other medications were recorded. Seizures were characterized according to the 2015 International Veterinary Epilepsy Task Force [19], with general epileptic seizures consisting of convulsive seizures with an alteration of consciousness. Only generalized convulsive epileptic seizures were evaluated

because electroencephalographic (EEG) confirmation of non-convulsive seizure activity was unavailable. Presence of any additional neurologic signs that could be indicative of partial seizures (motor, autonomic, behavioral) were also noted. Samples were collected based on the request of the attending veterinarian when medical concerns were noted. When no clinical concerns were noted, PB serum monitoring was scheduled every 4–8 weeks, primarily as a 24-h trough. Daily husbandry records were reviewed for the same time period, where any behavior changes were recorded periodically across approximately 12 h of each day. Feeding records were evaluated to assess total daily food consumption. Periods of prolonged anorexia were defined as complete inappetence with no food intake lasting longer than 48 h. During these periods, no PB was administered PO. Occasionally, an IM injection of PB was administered at the same dose as the prescribed PO dose during a period of prolonged anorexia. Administration of IM PB was dependent on recent PB serum concentrations and clinical judgment. Time points were excluded from PB dose and concentration analysis when PB was given as an IM injection within 7 days prior to a blood sample. When neither PO or IM PB was administered, either during a period of prolonged anorexia or when a previous serum concentration was above a desired level, a blood sample would reflect a time point longer than 24 h after oral administration. These time points were evaluated in association with other blood analyses described below, but were omitted from PB dose and PB serum concentration analysis. To evaluate total daily dose recommendations and therapeutic and ideal PB serum concentrations, only trough PB serum concentrations from blood samples collected 24 h after the last oral dose were included.

PB serum concentrations were measured at a reference laboratory (Idexx Laboratories, Westbrook, ME, USA). Routine complete blood count and serum biochemical analysis was performed from the same sample at the time of serum PB monitoring. To screen for evidence of potential hepatotoxicity associated with PB administration, alanine aminotransferase (ALT), aspartate transaminase (AST), alkaline phosphatase (ALP), and gamma-glutamyl transferase (GGT) were evaluated based on the distribution of tissue enzymes in this species [20]. Because tissue activity of transaminases such as ALT is also high in cardiac and skeletal muscle in CSL, creatine kinase (CK) was evaluated to differentiate between hepatic and muscle lesions; e.g., an elevation in ALT alone was likely to be hepatic in origin, while elevations in both ALT and CK were more likely to be from a muscle source. Transaminase and CK elevations were considered of clinical significance at greater than two times the upper limit of the reference range [21], based on previous studies of PB hepatotoxicity [22].

## 3. Results

An overview of the sample evaluation process and relevant results are presented in Figure 1. Blood samples were collected at 63 time points, ranging from 2 to 2442 h after the last oral dose. A complete blood count and serum biochemical analysis was paired with PB serum concentration for 49 of 63 time points. PB was administered PO for a total of 52 time points. Excluding time points when IM PB was administered within 7 days prior to sampling ($n = 11$), peak samples collected <4 h after dosing ($n = 3$), and trough samples collected >28 h after dosing ($n = 14$), thirty-five time points were trough samples collected 24 h after dosing. These 35 time points were used in evaluating the PB therapeutic range for this animal.

Generalized seizure control was achieved when serum PB concentrations were above 18.0 µg/mL. A total of 17 generalized convulsive seizures were observed during the study period. Sedation was noted in the medical records at time points when PB levels were above 35.0 µg/mL. Marked sedation and ataxia was noted during a two-week period when PB levels were between 58.0 and 68.0 µg/mL, resulting from a total daily dose between 2.6 and 3.4 mg/kg. For this animal, 18–35 µg/mL was the observed therapeutic range, with a mean PB concentration of 23.8 µg/mL ($\pm$5.8) resulting from a mean total daily dose of 1.5 mg/kg ($\pm$0.3). A proposed narrower, ideal range of 20–30 µg/mL was observed with the same mean total daily dose of 1.5 mg/kg ($\pm$0.3), resulting in a mean PB concentration of 24.8 µg/mL ($\pm$3.6).

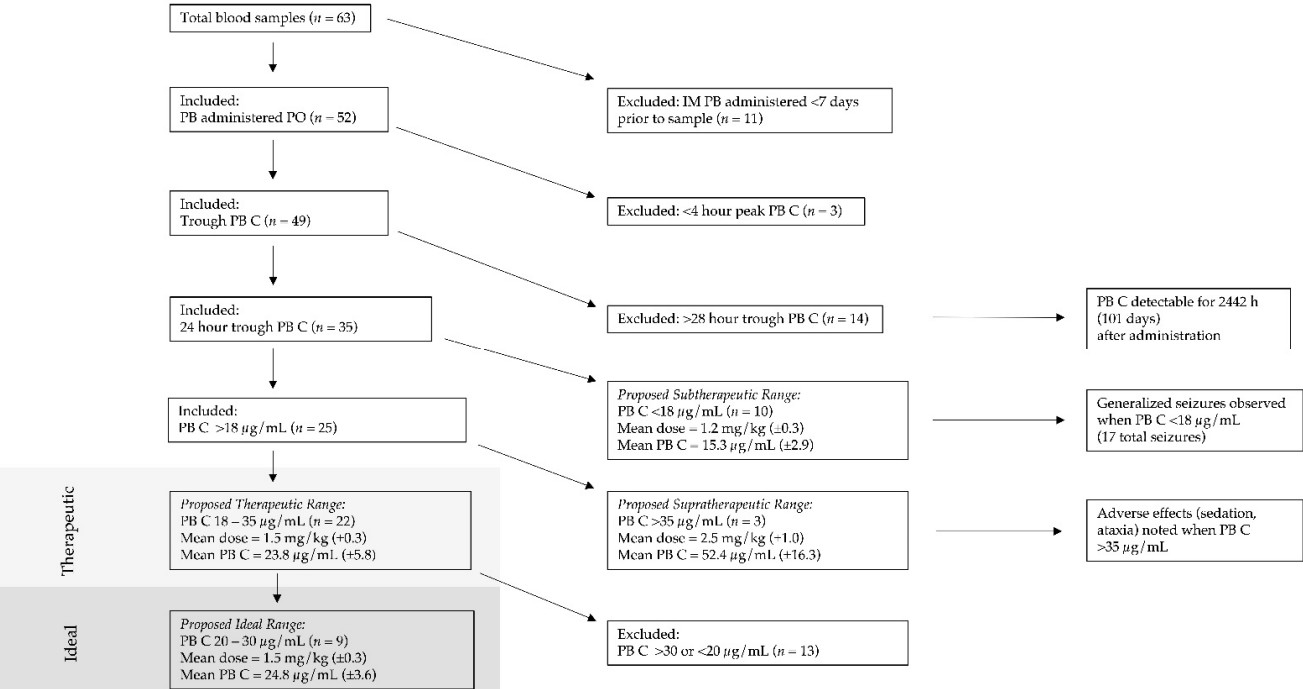

**Figure 1.** Schematic representation of sample evaluation towards recommended PB doses and both therapeutic and ideal PB serum concentrations. PB = phenobarbital; PO = per os (oral); IM = intramuscular; PB C = phenobarbital concentration. Dose refers to total daily dose.

Ten 24-h trough samples with a mean total daily dose of 1.2 mg/kg ($\pm$0.3) resulted in subtherapeutic PB concentrations (mean 15.3 $\mu$g/mL ($\pm$2.9)). Three 24-h trough samples with a mean total daily dose of 2.5 mg/kg ($\pm$1.0) resulted in supratherapeutic PB concentrations (mean 52.4 $\mu$g/mL ($\pm$16.3)).

Once daily (SID) dosing was compared with dosing divided into two doses (BID) for total daily doses ranging from 1.3 to 2.5 mg/kg, that could be reasonably assumed to fall within the therapeutic range. For 24-h trough samples, a mean SID dose ($n$ = 17) of 1.5 mg/kg ($\pm$0.2) resulted in a mean PB concentration of 22.5 $\mu$g/mL ($\pm$6.0). A mean BID dose ($n$ = 7) of 1.6 mg/kg ($\pm$0.3) resulted in a mean PB concentration of 22.8 $\mu$g/mL ($\pm$7.0).

In July 2019, after receiving 35 consecutive days of 2.6 mg/kg PO SID, the highest PB serum concentration (68 $\mu$g/mL) observed during the study period was recorded. After this time point, PB was discontinued for 101 days to allow PB concentrations to fall and while alternative AEDs were investigated. PB serum levels remained above 30 $\mu$g/mL for more than 500 h (21 days), and were detectable as far as 2442 h (101 days) after administration (Figure 2).

Xenotransplantation was performed in October 2020. The mean pre-surgery PB dose ($n$ = 18) was 1.6 mg/kg ($\pm$0.7), compared with a mean post-surgery PB dose ($n$ = 17) of 1.5 mg/kg ($\pm$0.3). The mean pre-surgery PB concentration was 24.2 $\mu$g/mL ($\pm$15.0), compared with a mean post-surgery PB concentration of 24.0 $\mu$g/mL ($\pm$6.4)

Between April and October 2020, there were four periods of time where the animal experienced prolonged anorexia, with a mean duration of 7.8 days (range 2–17 days). During this time, PB was administered as an IM injection twelve times. During this six-month period, only 19% (3/16) of PB serum concentrations were between 20 and 30 $\mu$g/mL, compared with 31% (10/32) of time points throughout the rest of the study period. Serum PB levels increased in three of the four periods of anorexia (range 9% to 29% increase) and decreased by 17% during one period of anorexia. Unexpectedly, when normal food intake resumed, PB levels dropped dramatically at the next sample timepoint, with a mean decrease of 27% (range 16% to 47%).

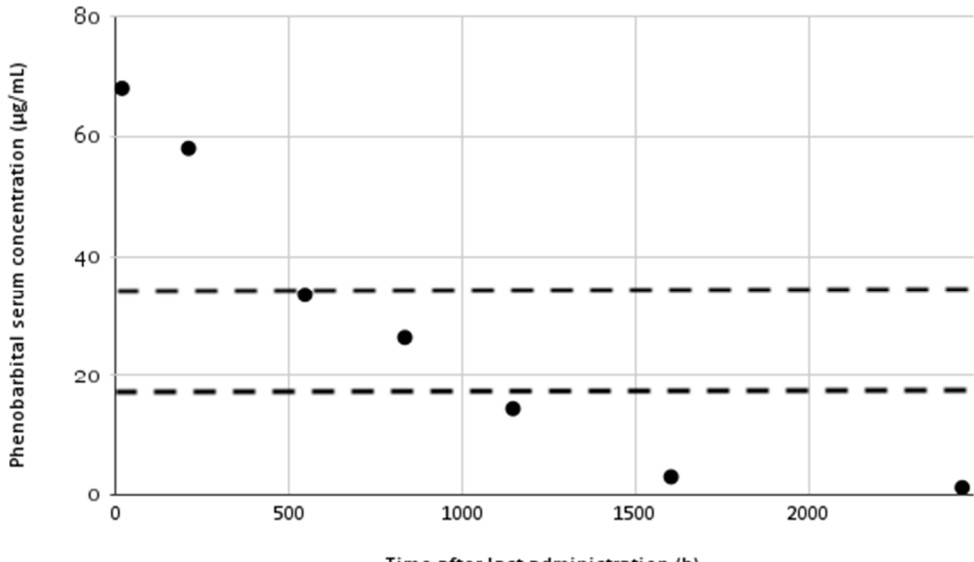

**Figure 2.** Observed serum phenobarbital concentrations versus time. The animal received 2.6 mg/kg PO SID PB for 35 days prior to the first time point, and no PB after that point. The dotted lines represent the proposed therapeutic range for this animal.

Two of 49 time points had elevations in ALT greater than twice the upper limit of the reference range (102 and 109 U/L, reference range 35–47 U/L). At one time point, a concurrent elevation in CK and AST was suggestive of a muscle source for the elevation. At the second time point, the PB level was subtherapeutic and not believed to be related to the elevated ALT concentration. No other clinically significant elevations in transaminases were observed at other time points.

## 4. Discussion

In this animal, seizures were noted at PB levels below 18.0 µg/mL, and adverse clinical signs of sedation and ataxia were noted at PB levels above 35.0 µg/mL. In dogs and cats, a therapeutic range of 15.0–56.0 µg/mL [23] is supplemented by the reference laboratory with a narrower 'ideal' range of 20.0–30.0 µg/mL (Idexx Laboratories). Aiming for the reference laboratory ideal range was more appropriate for this animal, particularly given its challenges with anorexia and consistent medication administration, and how long a level above therapeutic range persisted over time.

Human mean plasma half-life in adults is 79 h [24]. The data from this CSL, with a very slow terminal decrease in plasma concentrations, suggest the existence of a deep tissue compartment, and likely a half-life substantially longer than in humans or other veterinary species. This is supported by the fact that plasma concentrations were very similar between SID versus BID dosing, which is similar to what is reported in humans [25]. Statistical tests were not performed given that this was a single animal, but differences between SID and BID dosing were unlikely to have been clinically significant. SID dosing may assist with compliance, particularly in animals with a reduced appetite. SID dosing is likely appropriate for this species, although individual variation in drug absorption may exist, and more frequent administration may be required in cases where seizures are not well-controlled.

In dogs, PB can reportedly cause liver injury [26,27]. No clinically significant hepatic enzyme elevations were observed in this case or have been mentioned in any of the reports of PB use in this species (Table 1), and these findings support a low incidence of hepatic enzyme induction by PB in California sea lions. This assessment was only based on biochemical changes and not gross or microscopic hepatic changes. In other species, PB can

elicit gross liver enlargement, hepatocyte hypertrophy, and induced expression of drug-metabolizing enzymes [28]. Hepatic damage is rare in other species and thus conclusions in CSLs cannot be made from one animal, but long-term PB administration appears safe at doses that produce serum concentrations within these ranges.

Administration of an AED is considered effective when a patient has a ≥50% reduction in the number of seizures during a given treatment period [23,29–31]. This standard of efficacy was not used in this case because multiple confounding factors prevented a meaningful assessment, including the role of the xenotransplant in preventing seizures, and that the ability to maintain steady blood levels was highly impacted by behavioral changes that resulted in anorexia and an inability to consistently administer AEDs. Phenobarbital causes physical dependence, and rapid withdrawal of the drug itself can cause seizures [32,33]. Furthermore, a 50% reduction in the number of seizures is less than desired in a marine mammal species. Because of their aquatic nature, seizures when in the water pose a risk of drowning, and acute drowning has been reported in animals with neurologic disease [5]. Near-complete eradication of generalized seizure activity is, by necessity, a more appropriate goal for aquatic species.

The time period between April and October 2020 was marked by a progressive clinical decline characterized by prolonged anorexia and increasing seizure frequency, including cluster seizures. Given restrictions in monitoring, without EEG or consistent 24 h monitoring outside of emergent clinical windows, it is unknown whether partial or subclinical seizures were present and contributing to this decline. This period did not have good seizure control, but it cannot be concluded that this lack of control was due to PB, as anorexia prevented consistent medication administration. Even during this time period, no generalized seizures were observed when PB concentrations were greater than 18 μg/mL. It is reasonable to conclude that there was good generalized seizure control with PB concentrations between 20 and 30 μg/mL when consistent administration allowed these concentrations to be reached.

This animal's quality of life was frequently assessed throughout the clinical course of treatment by multiple staff using the facility's questionnaire. Particularly during the six months of clinical decline, humane euthanasia was considered. While this animal experienced prolonged anorexia, it did not show other signs such as lethargy or discomfort that typically accompany systemic disease, and bloodwork did not indicate systemic abnormalities. With few exceptions, the animal remained responsive and engaged in its typical behaviors. Clinicians should ideally follow standardized behavior guidelines to assess quality of life, and should be aware that certain neurologic signs, such as cluster seizures and status epilepticus, are associated with poor outcomes in epileptic patients [34].

This was a retrospective study using data collected opportunistically from a complex case. It is unknown what effect the xenotransplant had, if any, on PB pharmacodynamics, though the procedure's desired effect was to restore cellular function and thus ideally reduce or eliminate the need for AEDs. Of note is the fact that while mean PB dose and serum concentrations did not vary between the pre- and post-operative periods, the range of PB concentrations was wider in the pre-surgical period compared with the post-surgical period. This may be partially explained by a near resolution of anorexia, and thus an improvement in dosing consistency; it may also reflect gains in knowledge over time of the relationship between PB dose and concentration for this animal. Similarly, the administration of diazepam is another potential confounding variable. The dose given for the majority of the study (0.05 mg/kg PO) is well below typical veterinary doses for seizure control (0.25–0.5 mg/kg PO) [35], and in addition, chronic PB therapy reduces peak benzodiazapine concentrations [36], making it less likely that this dose of diazepam had an impactful effect on seizure control. Nonetheless, it is possible that the use of diazepam artificially lowered the PB concentration required for seizure control.

Prolonged anorexia was associated with relatively small changes in PB levels during the actual period without food consumption, although the resumption of normal food intake resulted in a rapid drop in PB levels. The mechanism behind these observed changes

is unknown. Phenobarbital is a lipophilic molecule. Lipids provide 80–95% of the energetic demand during fasting periods that are part of a pinniped's normal life history [37], and transfer of PB from fat to serum may occur during lipid mobilization. In addition, fasting differentially affects cytochrome P450-mediated drug metabolism [38], which may change how rapidly PB is cleared. Changes in pharmacokinetic parameters have been reported in dogs with differing body composition and diet [39]. In marine mammals, seasonal changes such as rut associated with breeding season in intact animals or events such as clinical disease that cause dramatic changes in body weight should be closely monitored. Drug dosage should be re-evaluated if there are substantial changes in diet, body weight or body composition. Clinicians should be aware of the potential for serum PB drop following resumption of normal diet, which may require a temporary dose increase to avoid subtherapeutic drug levels and a subsequent seizure.

Reported adverse effects of PB include sedation, ataxia, weight loss and behavioral changes [35]. Because these adverse effects overlapped with the signs associated with this animal's neurological disease, it is difficult to conclude when side effects were an adverse effect of PB administration and when they were due to the neurological disease. Adverse effects did appear to be dose-dependent, with dramatic sedation and ataxia reported at PB levels above 58.0 μg/mL. Anorexia, weight loss, and behavioral changes appeared to be associated with this animal's structural brain disease, as it was noted at a wide variety of PB levels, and improved following xenotransplantation. The procedure led to a greater than 90% reduction in seizures and a stabilization of weight and food consumption [18]. This clinical progression towards refractory epilepsy is supported by other reports of sea lions with DA toxicosis where PB was not ultimately effective in controlling seizures [7]. Still, although multiple factors may have influenced the clinical signs observed, clinicians should be aware of the potential for adverse effects of PB itself, particularly for PB serum levels above 35.0 μg/mL.

For this animal, a serum PB target range of 20.0–30.0 μg/mL was achievable with a dose of 1.5 mg/kg SID followed by therapeutic monitoring, and this is a reasonable starting point recommendation for clinicians treating this species. This initial research seeks to guide clinicians managing epilepsy in this species, but future pharmacokinetic analyses should be properly powered and prospectively designed. Ideal long-term seizure control may be difficult to achieve with anti-epileptic drugs such as PB alone. With the health threats from marine biotoxin-producing harmful algal blooms on the rise, further research is needed to make novel options useful for clinical management of biotoxin-related neurologic disease in this aquatic species.

**Author Contributions:** All authors had significant contributions to this study. Conceptualization, C.A.S., G.S., R.A.N. and D.P.; methodology, C.A.S., G.S. and D.P.; validation, C.A.S. and R.A.N.; formal analysis, C.A.S.; investigation, C.A.S. and R.A.N.; resources, C.A.S., G.S. and D.P.; data curation, C.A.S.; writing—original draft preparation, C.A.S.; writing—review and editing, C.A.S., G.S., R.A.N. and D.P.; visualization, C.A.S.; supervision, C.A.S.; project administration, C.A.S., G.S. and D.P. All authors have read and agreed to the published version of the manuscript.

**Funding:** This research received no external funding.

**Institutional Review Board Statement:** Ethical review and approval were waived for this study because this was a retrospective analysis of a dataset. The data collected were all associated with the medical care of the animal, were collected as part of routine diagnosis and treatment, and as such does not require ethics committee oversight as per the American Veterinary Medical Association. Testing blood is essential, is done without fail as part of veterinary care, and is not an add-on for research purposes. This research does not report on the use of experimental protocols; the diagnostics reported were based on the existing literature in this and other species. This clinical case report follows a best practice of veterinary care.

**Data Availability Statement:** Restrictions apply to the availability of these data. The data presented in this study are available on request from the corresponding author with the permission of Six Flags Discovery Kingdom.

**Acknowledgments:** The authors wish to thank Kelly Goulet, Dawn Robles and Joy Middleton for their role in sample collection and processing, and to Six Flags Discovery Kingdom's dedicated veterinary staff and training and husbandry teams for providing the highest standard of care to this sea lion, and to all of the animals in their care. In addition, the authors wish to thank the two anonymous reviewers for their feedback that strengthened the final manuscript.

**Conflicts of Interest:** The authors declare no conflict of interest.

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
