# Peer review of "Clinical Observations Associated with Phenobarbital Serum Monitoring to Manage Epilepsy in a California Sea Lion with Domoic Acid Toxicosis"

_2673-1924, doi:10.3390/oceans3030023_

Round 1

Reviewer 1 Report

Review report «Clinical observations associated with phenobarbital serum monitoring to manage epilepsy in a California sea lion with domoic acid toxicosis»

Abstract:

Line 17 and 18: “serum levels remained detectable as far as 101 days after administration….” This makes me think you stopped giving phenobarbital, and continued monitoring? Please explain more precisely.

Line 21 and 22: results based on one single animal can’t be used as a recommendation. I would re-word. For instance “and this is a reasonable recommended concentration as an initial dose for clinicians treating subadult male CSLs”

Line 26: key words that are already in the title are less useful – just a tip

Introduction: very short, I guess you are struggling with word limit? The second paragraph is very condensed, and I suggest you outline a bit more about rehabilitation and long term care facilities, which isn’t very common outside of the USA (or even outside of California). This doesn’t have to be long winded, but make sure you separate between free-ranging animals, animals in captive facilities et.c.

Table 1:

PB serum conc : is this the target dose, or whatever dose the study reported? Should be clarified.

Number of animals: “multiple” is not sufficient. Get the correct sample size.

Number of serum samples does not give much information. I’d suggest changing to time period animal was monitored, or delete this column.

M and M:

Line 56: are 8 year-olds still considered sub adults?

63-66: please detail how these drugs were administered PO.

95 – 99: statistics on N = 1? You have many data points, but from the same animal, with differing treatment schemes. Doing a paired t-test on several samples from the same animals, must be interpreted extremely cautiously, and can’t be assumed as a significant test for the population, because you are only testing one animal. Please elaborate on this. Another problem is that the animal had surgery to restore hippocampal activity halfway through the study period. Did you include only samples before or after surgery, since this procedure must be seen a major bias for the continuity of the dataset? This goes for both the SID vs BID testing, but also the final recommended SID dose. Is this based on data collected before or after surgery? Did the surgery change the result? Another possible problem is the other medication this animal was on. Barbiturates for seizure and behavioral control. Was this accounted for?

Results:

102-103: was treatment stopped at some point? Otherwise, blood samples could nevcer be sampled later than 24 hrs since last SID treatment?

108: if you decide to keep the statistical analyses, please report DF and T as well.

108 – 110 + figure 1: I still don’t understand this measurement. Were animals not dosed every day? I assume the rigidness of word count may have edited away the explanation for this figure and text?

Figure 2: not sure this figure adds much that’s not already carefully outlined in the text.

Discussion:

Line 144: there is a big difference between toxicity and sedation. In the abstract you state that animals became more sedated at levels above 35 µg/mL, which one is correct?

Overall: the discussion is well written and is carefully not over-interpreting results collected from N = 1. I think a paragraph, or at the very least a few sentences could be added discussing the ethics of continuing care for this animal, which is going through prolonged anorexic periods, making the epilepsy harder to control (and probably increasing seizures during these periods?). When do the authors recommend other clinicians to consider euthanasia?

Author Response

Thank you very much for your review. Please see the attachment for detailed point-by-point responses. 

Reviewer 2 Report

The manuscript is well-written with really good information regarding serum phenobarbital levels correlated to phenobarbital dosing that is absent from current literature. This information is extremely important to the veterinary community managing epileptic marine mammals, specifically California sea lions. 

Major revisions/comments:

Can the authors better define seizures and thus seizure control by phenobarbital? There are many instances in the manuscript where the animal had behavioral abnormalities and anorexia that could have been secondary to subclinical seizures that may have gone undetected without EEG or close monitoring and thus was not well-controlled on the recommended phenobarbital dose. Along the same line, can the authors describe if there was 24 hour monitoring on this animal? If not, could seizures that occurred overnight been missed on the proposed phenobarbital dosing when the authors are stating that the animal was well-controlled?

Is there a reason the authors did not evaluate other transaminases (AST, GGT, ALKP) for evidence of hepatocellular injury. I agree that ALT is the most specific in this species for liver injury, however GGT has also been elevated in severe hepatic disease in this species despite its specificity for the kidneys. The evidence to support no hepatocellular injury secondary to phenobarbital administration in this animal would be more complete if all transaminases were evaluated.

How did the authors deal with IM phenobarbital injections and the serum levels of PB? Can the authors better describe how they kept the injectable samples out of the analysis? Specifically during the prolonged periods of anorexia, was the animal not receiving phenobarbital at all during this time? What was the time frame used between injectable phenobarbital doses and serum samples analysed for phenobarbital levels to make sure the injectable doses were not included since phenobarbital has such a prolonged half life in this species?

Under Case Definition: this animal actually had a grand mal seizure in rehabilitation after stranding 3 times which was part of the reason for being deemed unreleasable. The animal also had a normal MRI (normal hippocampus) in rehabilitation. Information would be good for readers to know in general to better understand potential disease progression and show that hippocampal atrophy can occur later despite seizures being present. 

Did the authors use any other diagnostics to rule out hepatocellular injuries common in dogs like liver enlargement in this case? Ultrasound? Biopsy? Others? If no, can the authors be more clear that their evaluation in liver damage was purely based on biochemical changes and not gross or microscopic changes?

Paragraph starting at line 194 adequately describes that the behavioral changes in this animal including the prolonged periods of anorexia could have been due to its underlying neurological disease. However, since this animal had significant behavioral changes while on the recommended dose of phenobarbital proposed by the authors, is it fair to say that pheno levels in this animal between 20-30 mcg/mL adequately controlled its seizures/neurological disease?

Could the authors state what the phenobarbital sampling schedule was for this animal? Opportunistic and if so when/under what circumstances were levels typically requested? Or was it a set sampling schedule to monitor levels on a routine basis?

Minor revisions:

Add in dose frequency for phenobarbital dosing on line 64

Were the SID doses of phenobarbital doubled? Could the authors better explain the dosing to make sure readers know if the SID schedule was equal in dosing to the BID schedule?

Could the authors add in at what dose this animal was on when the phenobarbital levels increased to 58-68 mcg/mL? Agree that the clinical signs were most likely from phenobarbital toxicity, but can the authors completely rule out that animal's clinical signs were not due to the neurological disease?

Change phenobarbital to PB on line 165 to make sure abbreviations are consistent throughout manuscript.

Add in p-value and degrees of freedom from t-test in results for significance between SID and BID dosing.

Author Response

Thank you very much for your helpful comments. Please see the attachment for detailed point-by-point responses. 

Round 2

Author Response

Please see the attachment for a point-by-point response. 

Reviewer 2 Report

I appreciate the author's changes and responses to first version of the manuscript. Their changes make the manuscript and story of this animal much more complete and a good resource for readers who may also be managing animals similar to this case. 

I have one further comment/confusion with this version of the manuscript that may just be due to the track changes that are on and is related to the figures in the manuscript. For Figure 1, I see two different figures. The top one appears more complete with multiple sampling time points while the bottom figure appears to be a partial version of the top figure. Both axes are labeled the same with just a difference in mcg/mL vs. ug/mL on the y-axis. If this is not a mistake, could the authors please describe in the Figure description what each figure is depicting so it is more clear as well as label one figure with A and one figure with B or some other descriptor to make sure readers know what they are looking at?

Similarly, I still see Figure 2 in the manuscript, however the description for the figure is crossed out in track changes as if it has been deleted. I'm assuming the figure is meant to be removed, however if it will still be in the manuscript, could the authors make sure to have a figure description to go along with it?

Author Response

(The authors gave the same response as above.)

Round 3

Reviewer 1 Report

At this point I think the manuscript is ready for publication. I do however think that figure 1 can be deleted. If you insist on keeping it, it should be edited for clarity (right now it is messy). It can be improved by removing excluded time points and removing the headline "included time points" (which then presents the total N, and 11 of these samples are excluded, giving the headline included timepoints ambiguity.) I think the recommendations are clear in the text, and the figure can be skipped, or simplified.